# Long-Term Response to Tyrosine Kinase Inhibitors for Metastatic Renal Cell Carcinoma

**DOI:** 10.3390/biomedicines10102444

**Published:** 2022-09-30

**Authors:** Martina Catalano, Ugo De Giorgi, Marco Maruzzo, Davide Bimbatti, Sebastiano Buti, Giulia Mazzaschi, Giuseppe Procopio, Matteo Santoni, Luca Galli, Raffaele Conca, Laura Doni, Lorenzo Antonuzzo, Giandomenico Roviello

**Affiliations:** 1Department of Health Sciences, University of Florence, 50134 Florence, Italy; 2IRCCS Istituto Romagnolo per lo Studio dei Tumori (IRST) “Dino Amadori”, 47014 Meldola, Italy; 3Oncology Unit 1, Istituto Oncologico Veneto IOV-IRCCS, 35128 Padua, Italy; 4Department of Medicine and Surgery, University of Parma, 43121 Parma, Italy; 5Department of Medical Oncology, Fondazione IRCCS Istituto Nazionale dei Tumori di Milano, 20133 Milan, Italy; 6Oncology Unit, Macerata Hospital, 62100 Macerata, Italy; 7Medical Oncology Unit 2, Azienda Ospedaliero-Universitaria Pisana, 56126 Pisa, Italy; 8Unit of Medical Oncology, Department of Onco-Hematology, Centro di Riferimento Oncologico della Basilicata (IRCCS-CROB), 85028 Rionero in Vulture, Italy; 9Clinical Oncology Unit, Careggi University Hospital, 50134 Florence, Italy; 10Department of Experimental and Clinical Medicine, University of Florence, 50134 Florence, Italy; 11Department of Health Sciences, Section of Clinical Pharmacology and Oncology, University of Florence, Viale Pieraccini 6, 50139 Florence, Italy

**Keywords:** renal cell carcinoma, TKI, good risk, poor risk

## Abstract

**Background:** Tyrosine kinase inhibitors (TKIs) prolong progression-free survival (PFS) in patients with metastatic renal cell carcinoma (mRCC), some of which may achieve long-term responses. Herein, we report clinical and pathological characteristics of patients who achieved long-term responses during first-line TKI treatment. **Methods:** Patients receiving TKI as first-line therapy from January 2010 to December 2017 in seven Italian Oncology Centers were reviewed. Sixty-six patients were considered as long-term responders, as they remained progression-free for 36 months or more during TKI treatment. A logistic regression model was performed to evaluate the effect of each clinical-pathological variable on the probability of responding long-term. **Results:** A total of 335 patients with a median age of 66 years were included in the analysis. The median PFS and overall survival among the long-term responders was 70 and 106 months, respectively. At a landmark PFS analysis performed 36 months after the start of treatment, the median PFS was 34 months. Multivariate analysis from all patients identified previous nephrectomy, Eastern Cooperative Oncology Group Performance Status (ECOG PS) < 1, and lack of liver metastasis as favorable prognostic factors for long-term response. Female gender and lack of liver metastasis positively correlated with long-term responses in favorable-risk-score population, as well as ECOG PS < 1 in intermediate-poor risk score population. **Patients Summary:** Previous surgery, clinical condition, and lack of liver metastasis may predict long-term responses to tyrosine kinase inhibitors. **Conclusions:** TKIs can lead to a long-term response in a subset of patients with metastatic RCC. Previous nephrectomy, optimal performance status (ECOG PS = 0), and lack of liver metastasis may predict long-term responses.

## 1. Introduction

Renal cell carcinoma (RCC) accounts for over 90% of cancers in the kidney with clear cell RCC the most common histological and molecular subtype [1]. Localized RCC can be successfully managed with nephrectomy, whereas metastatic RCC is highly resistant to conventional chemotherapy. Increased understanding of the biological characteristics of RCC has enabled the development and implementation of new therapeutic strategies in the field of metastatic RCC. Immune checkpoint inhibitors, commonly in association with targeted agents including cabozantinib, bevacizumab, and axitinib, or in combination with each other, have become the standard of care in first-line treatment [2]. Patient outcome is significantly increased (compared to monotherapy) with tyrosine kinase inhibitor (TKI), in any case at the expense of a greater toxicity [3,4,5,6]. However, for several years sunitinib and pazopanib, multitargeted TKIs, have improved the possibility of metastatic RCCs achieving response rates between 30% and 50%, a median progression-free survival (PFS) of 8.4–11 months, and overall survival (OS) of 26.4–28.4 months [7,8]. In this study, we report a multicenter experience where a subset of patients achieves long-term responses, defined as patients with mRCC remaining progression-free for 36 months or more with TKI therapy. To describe the clinical and pathological characteristics of these patients, we performed a retrospective analysis of all patients with metastatic RCC treated with sunitinib and pazopanib in several Italian Oncology Centers. We report on clinical-pathological features, treatment outcomes, and survival for long-term responders receiving TKI as first-line setting.

## 2. Materials and Methods

### 2.1. Study Population and Schedule Treatment

Three hundred thirty-five patients with metastatic RCC who received sunitinib or pazopanib between January 2010 and December 2017 in seven Italian Oncology Centers were included in this retrospective study. Inclusion criteria included age > 18 years and histologically confirmed RCC with completion of at least one cycle of treatment. The exclusion criteria consisted of patients with cardiovascular disease, hepatic or renal insufficiency, infectious disease, other malignancies, or those previously treated. Patient demographics and clinical features at study entry were recorded and those with incomplete clinical data were excluded. All patients received target therapy with sunitinib 50 mg for four consecutive weeks and a two week pause, or pazopanib 800 mg once a day continuously up to intolerance or progression of disease. Written informed consent was obtained from each patient before starting treatment. This study was approved by the Comitato Etico Regionale for clinical experimentation of Toscana Region (Italy) area Vasta Centro Section, (Tuscany section area Vasta Centro, number: 14565_oss). The study was performed in accordance with the Declaration of Helsinki.

### 2.2. Outcome Assessment and Statistical Analysis

Tumor response evaluation was performed every three months, or earlier when clinically required, by spiral computed tomography according to Response Evaluation Criteria in Solid Tumors (RECIST) version 1.1 [9]. PFS was evaluated as time from the beginning of the treatment to the date of the disease progression. OS was evaluated as the time from TKI regimen start to death from any cause or the date of the last follow-up visit. Kaplan–Meier methodology was used to estimate both PFS and OS during follow-up. PFS analysis was performed at a landmark timepoint 36 months after the start of therapy to avoid bias of the resulting Kaplan–Meier survival estimates. Landmark analysis was performed to estimate the additional progression-free duration among patients who were able to continue TKI therapy without progression for 36 months (long-term responders). To compare the categorical baseline characteristics distribution of the long and short-term responders, the *χ*^2^ test was used. The variables were described as median and range for quantitative data and by numbers and percentages for qualitative data. A logistic regression model was performed on all patients regardless risk class and based on risk score (favorable or intermediate-poor) to evaluate the effect of each clinical-pathological variable on the probability of responding long-term. A *p* < 0.05 score was considered statistically significant. Statistical analysis was performed using STATA version 9.1.

## 3. Results

### 3.1. Patient’s Features According to Duration of Response

Between January 2010 and December 2017 a total of 368 patients diagnosed with metastatic RCC and treated with the TKI at the front-line were retrospectively investigated. Of these, 33 patients receiving cabozantinib were excluded. Overall, 335 patients were enrolled in our retrospective clinical trial. Sixty-six patients (19.7%) received TKIb for ≥36 months and were identified as long-term responders. Baseline characteristics for the long-term and short-term responders are shown in Table 1. Both groups had similarities in median age, sex, and histology. Clear cell carcinoma was the predominant histologic type, accounting for 87.8% of all patients. Nephrectomy, before initiating systemic therapy, had been performed in over 80% of all patients with statistically significant predominance in the long-term responders (*p* < 0.01). The distribution of patients by International Metastatic RCC Database Consortium (IMDC) risk group showed a higher percentage of patients with intermediate-poor risk status in the short-term response group (68.4% vs. 53.0%; *p* = 0.02). The proportion of patients with liver metastases was higher in the short-term responders than in the long-term responders (17.5% vs. 6.1%; *p* = 0.02). Finally, no statistically significant differences were observed for first-line TKI and number of therapy lines >1 after TKI in the two groups.

### 3.2. Patients Features According to Duration of Response and IMDC Score Risk

One hundred sixteen patients were considered as at a favorable risk according to IMDC classification (Table 2). Of these, 66 (56.9%) received sunitinib as first-line therapy and the remaining ones pazopanib. Thirty-one (26.7%) patients achieved a PFS ≥ 36 months and no statistically significant difference were observed with respect to the TKI used. Among metastatic sites, only liver involvement showed a significant difference between long-term (3.2%) and short-term (20.2%) responders (*p* = 0.04). Others baseline characteristics were similar in the two groups. A smaller percentage of patients (12.9%) received more than 1 line of therapy after TKI, in the long responder cohort compared with the short responder group (34.1%) (*p* = 0.03).

Two hundred nineteen patients were classified as IMDC intermediate-poor risk (Table 3). In this subgroup, only the 16.0% of patients achieved a PFS ≥ 36 months. Clinical and demographic features were substantially balanced in the two groups without significant differences, except for ECOG PS and previous nephrectomy. Indeed, a higher percentage of patients with PSF < 36 months (60.3%) had ECOG PS ≥ 1 compared with long-term responders (37.1%) (*p* < 0.01). Regarding surgery treatment, 94.1% of patients with PFS ≥ 36 month and 77.7% of patients with PFS < 36 months had been subjected to previous nephrectomy (*p* = 0.02).

### 3.3. Response Rate and Outcome Efficacy

A response rate (RR) of 37.1% and a disease control rate (DCR) of 81.1% were recorded for the total population with a significant statistically difference between long- and short-term responses (*p* < 0.01). Similarly, a higher RR and DCR were observed in patients achieving a PFS ≥ 36 months compared with patients with PFS < 36 months, in favorable (*p* = 0.05; *p* = 0.03) and intermediate-poor risks (*p* = 0.05; *p* < 0.01), respectively (Appendix A). Median PFS and OS were assessed in all patients and resulted in 13 (95% CI, 11–15) and 48 (95% CI, 39–60) months, respectively. In long-term responders, median PFS was 70 months versus 10 months in patients achieving PFS < 36 months (Appendix A). Median PFS data in IMDC favorable and intermediate-poor risk population are reported in Appendix A. In patients with PFS ≥ 36 months, median OS was 106 (95% CI, 93-not reached [NR]) compared with 33 (95% CI, 24–39) months in patients with PFS < 36 months (HR, 0.20; 95% CI, 0.12–0.32; *p* < 0.01) (Appendix A and Figure 1).

Among long-term responders with an IMDC favorable risk, mOS was 105 (95% CI, 70-NR) versus 66 (95% CI, 40–75) months of patients with PFS < 36 months (HR, 0.30, 95% CI, 0.14–0.62; *p* < 0.01). Similarly, a longer mOS was recorded in intermediate-poor risk patients with PFS ≥ 36 months (112 months, 95% CI, 93-NR) compared to those with PFS < 36 months (23 months, 95% CI, 20–29) (HR, 0.17; 95%CI, 0.09–0.32; *p* < 0.01) (Appendix A and Figure 2).

At a landmark time of 36 months in the long-term responders, median PFS and OS was 34 (95% CI, 22–50) and 70 (95% CI, 57-NR) months, respectively (Figure 3).

### 3.4. Long-Term Response Predictors

Logistic regression analysis was performed on all 335 patients to assess the relationships between clinical-pathological variables and long-term responses. Risk variables assessed included age, gender, histology type, previous nephrectomy, ECOG PS, sarcomatoid feature, IMDC score, and metastatic site. The odds ratio (OR) estimate for each variable in the univariate and multivariate analysis is listed in Table 4.

Long-term responders had greater odds of having received previous nephrectomy (OR, 3.42; 95% CI, 1–11.63; *p* = 0.05), and a lower odds of having ECOG PS ≥ 1 (OR, 0.50; 95% CI, 0.30–0.85; *p* < 0.01) and liver metastases (OR, 0.32; 95% CI, 0.11–0.93; *p* = 0.04), compared with short-term responders. Univariate and multivariate analysis of the relationship between clinical-pathological variables with PFS ≥ 36 months in favorable and intermediate-poor risk patients are reported in Table 5 and Table 6.

Patients with PFS ≥ 36 months and IMDC favorable risks were less likely to be male (OR, 0.38; 95% CI, 0.15–0.95; *p* = 0.04) and to have liver metastasis (OR, 0.32; 95% CI, 0.11–0.93; *p* = 0.04) compared with PFS < 36 months patients. Among intermediate-poor risk patients, long-term responders had lower odds of having ECOG ≥ 1 (OR, 0.44; 95% CI, 0.21–0.94; *p* = 0.03) compared to short-term responders.

## 4. Discussion

Although no longer the standard of care at present, TKIs were the first-line choice therapy for metastatic RCC for several years. In the randomized phase III pivotal trial, sunitinib showed a longer PFS (11 versus 5 months; *p* < 0.05) and OS (26.4 versus 21.8 months; *p* = 0.05) as well as response rates (47% versus 12%; *p* < 0.05) compared with interferon alfa [10]. Successively, compared with sunitinib, pazopanib was non-inferior with respect to PFS (8.4 versus 9.5 months; HR, 1.05; 95% CI 8.3–10.9) and OS (28.4 versus 29.3 months, HR, 0.91; 95% CI, 0.76–1.08) [8]. In this retrospective study, we analyzed data collected from the medical records of 335 patients with mRCC who received sunitinib or pazopanib as first-line therapy. Our goal was to present the clinical and pathological characteristics, as well as survival outcomes, of patients considered as long-term responders that reached a PFS ≥36 months. Median OS was 48 months for the total population; 75 months in patients with a favorable risk and 33 months for those with an intermediate-poor risk. Notably, in long-term responders with an intermediate-poor risk, mOS was higher in long responders with a favorable risk (112 vs. 105 months). These survival data are larger than those reported in clinical practice studies, and may reflect the presence of considerable heterogeneity in clinical-pathological characteristics of patients [11,12,13,14,15]. We identified previous nephrectomy, ECOG PS < 1, and lack of liver metastases as factors associated with long-term responses in the study population. Variables associated with long-term response in IMDC favorable-risk patients included lack of liver metastases and female gender, whereas ECOG PS < 1 was associated with PFS ≥36 months in IMDC intermediate-poor risk patients (Figure 4). Results are consistent with data in the literature reporting a better overall survival, recurrence rates, and cancer specific survival in women than in men with RCC and a good performance status as a predictive factor of better response [16].

In our analysis, the ratio of patients remaining long-progression-free for 36 months or longer with TKI therapy was similar to that reported in a previous long-term response study of sunitinib and pazopanib with accordance to the general characteristics of patients [17,18]. Ana M. Molina et al. retrospectively identified a subgroup of long-term responders (18.9%), defined as patients remaining progression-free while receiving sunitinib for >18 months or achieving a complete response (CR) during treatment with sunitinib. In this study, patients received sunitinib for a median of 24.9 months, and a maximum duration of 73.9 months [17]. The objective response rate was 79%, with three patients achieving CR. On a landmark PFS analysis performed 18 months after start of treatment, median PFS was 17.4 months. Authors found that lack of bone metastases or lung metastases and favorable Memorial Sloan–Kettering Cancer Center (MSKCC) risk status were associated with long-term responses [17]. Similarly, Mustafa et al. reported a 19.3% proportion of patients remaining progression-free for 18 months or longer while on pazopanib [18]. Baseline patient characteristics associated with PFS benefit (>18 months), CR, and partial response (PR) included age < 65 years and favorable risk based on either IMDC or MSKCC criteria, although data for risk classifications were unavailable for about one third of patients [18]. Favorable or intermediate MSKCC risk has been identified as risk factor correlated with a greater likelihood of achieving CR with sunitinib treatment by Heng et al. [19], whereas other authors have not been able to identify clinical or biological parameters associated with long-term responses or complete remissions [20]. When we started the study, TKIs were considered the standard of care at the forefront of metastatic RCC patients in most countries [21,22]. Currently, immune checkpoint inhibitors, both combined with TKIs and with each other, have considerably changed the treatment paradigm of RCC [3,4,5,6], reserving the use of TKI monotherapy for limited cases [23,24]. However, in the favorable-risk population, combination therapies did not show a significant advantage in terms of OS over TKI alone, at the expense of a greater toxicity [3,4,5,6]. The presence of clinical-pathological variables, correlated with long-term responses, could support the best therapeutic choice in clinical practice, mainly in favorable-risk mRCC patients with low burden of disease, slowly progressing disease, and lack of unfavorable metastatic sites (liver or brain metastasis). In these patients, the use of TKI alone may still be justified. There are several limitations in our study. Firstly, the retrospective nature of data collection. Secondly, the incompleteness of the data relating to best response (CR or PR), and the lack of information on the duration of treatment, dose adjustment, and safety data. The strengths are represented by the large sample examined, the multicenter nature of the study, and the long time of progression-free disease considered despite the arbitrary choice of the cut off.

## 5. Conclusions

In summary, tyrosine kinase inhibitors achieve long-term responses in a subset of patients with metastatic RCC. Previous nephrectomy, ECOG PS < 1, and lack of liver metastasis may predict long-term response regardless of risk classification. Female gender and lack of metastatic site predict long-term response in the favorable-risk population; ECOG PS < 1 in intermediate-poor-risk population. Clinical-pathological features could favor optimal therapeutic first-line choice in mRCC patients.

## Figures and Tables

**Figure 1 biomedicines-10-02444-f001:**
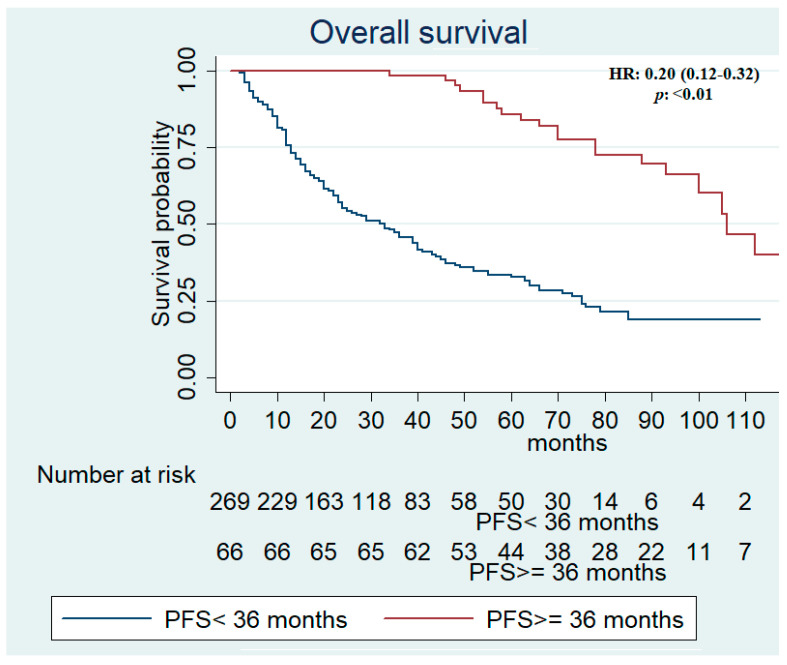
Kaplan–Meier survival estimate according to progression-free survival in all patients.

**Figure 2 biomedicines-10-02444-f002:**
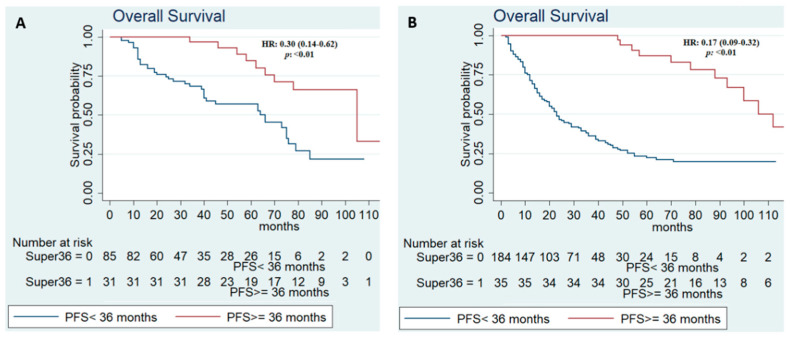
Kaplan–Meier survival estimate according to progression-free survival in all patients in good risk (**A**) and intermediate-poor risk (**B**).

**Figure 3 biomedicines-10-02444-f003:**
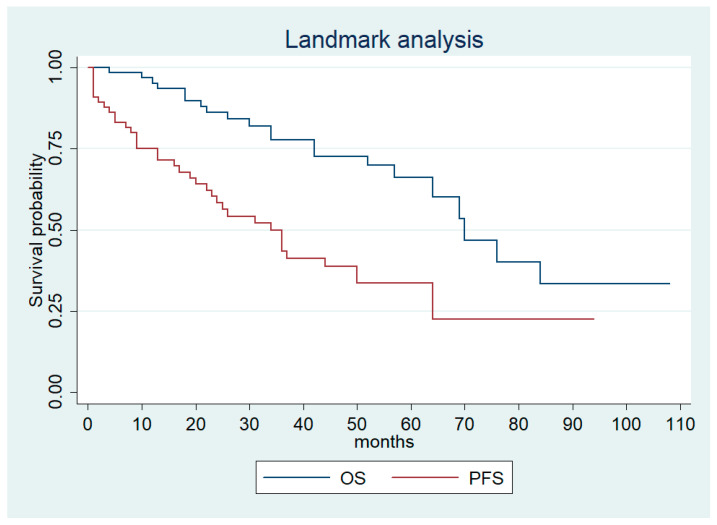
Landmark progression-free survival and overall survival for long-term responders.

**Figure 4 biomedicines-10-02444-f004:**
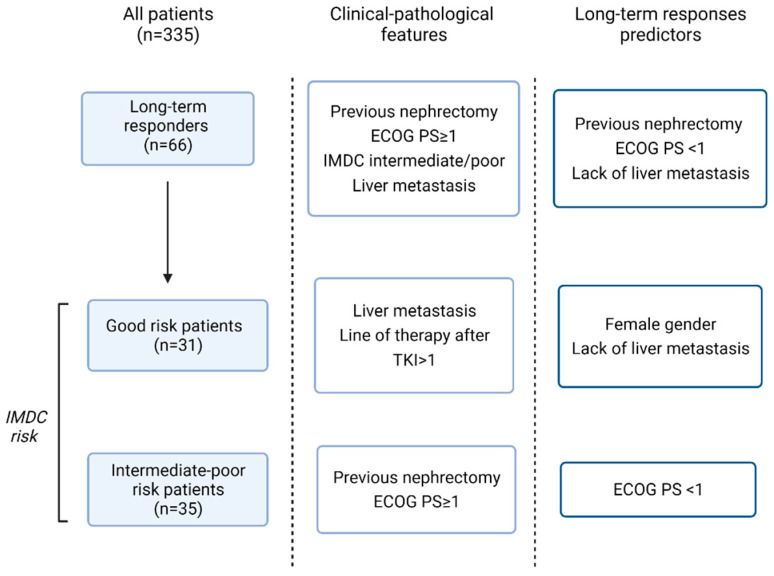
From left to right the flow chart shows: number of patients achieving progression-free survival (PFS) ≥ 36 months; the statistically significant difference between clinical-pathological features in patients with PFS ≥ 36 months or PFS < 36 months; the variables predicting long-term responses.

**Table 1 biomedicines-10-02444-t001:** Patients’ baseline characteristics according to duration of response to TKI.

	All	PFS ≥ 36 Months	PFS < 36 Months	*p*
	*N* = 335	*N* = 66 (19.7%)	*N* = 269 (80.3%)	
**Age**				0.9
Median (range)	66 (37–89)	67 (37–85)	66 (37–89)	
**Gender,***n* (%)				0.1
Male	240 (71.6)	43 (65.1)	197 (73.2)	
**Histology,***n* (%)				0.4
Clear-cell RCC	294 (87.8)	59 (89.4)	235 (87.4%)	
**Previous nephrectomy,***n* (%)				<0.01
Yes	286 (85.4%)	63 (95.4)	223 (82.9)	
**ECOG,***n* (%)				<0.01
≥1	170 (50.7)	22 (33.3)	148 (55.0)	
**Sarcomatoid feature**				0.1
Yes	14 (4.2)	0	14 (5.2)	
**IMDC score,***n* (%)				0.02
Intermediate-poor	219 (65.4)	35 (53.0)	184 (68.4)	
**Metastatic sites,***n* (%)				
Lung	218 (65.1)	40 (60.6)	178 (66.2)	0.2
Liver	51 (15.2)	4 (6.1)	47 (17.5)	0.02
Nodal	117 (34.9)	21 (31.8)	96 (35.7)	0.2
Bone	105 (31.3)	18 (27.3)	87 (32.3)	0.3
Brain	14 (4.2)	1 (1.5)	13 (4.8)	0.1
**First-Line Therapy,***n* (%)				0.4
Sunitinib	200 (59.7)	38 (57.5)	162 (60.2)	
Pazopanib	135 (40.3)	28 (42.5)	107 (39.8)	
**Line of therapy after TKI,***n* (%)				0.1
>1	102 (30.4)	15 (22.7)	87 (32.3)	

Percentages are expressed on column total. RCC: renal cell carcinoma, ECOG PS: Eastern Cooperative Oncology Group Performance Status; IMDC: *International Metastatic Renal Cell Carcinoma Database Consortium*; TKI: tyrosine kinase inhibitor; *p*: *p*-value.

**Table 2 biomedicines-10-02444-t002:** Patients’ baseline characteristics according to duration of response to TKI in favorable-risk patients.

	All	PFS ≥ 36 Month	PFS < 36 Months	*p*
	*N* = 116	*N* = 31 (26.7%)	*N* = 85 (73.3%)	
**Age**				0.3
Median (range)	69 (37–86)	71 (44–85)	69 (37–86)	
**Gender,***n* (%)				0.1
Male	83 (71.5)	18 (58.1)	75 (76.5)	
**Histology,***n* (%)				0.7
Clear-cell RCC	102 (87.9)	28 (90.3)	74 (87.1%)	
**Previous nephrectomy,***n* (%)				0.9
Yes	110 (94.8%)	30 (96.8)	80 (94.1)	
**ECOG,***n* (%)				0.2
≥1	46 (39.7)	9 (29.0)	37 (43.5)	
**Sarcomatoid feature**				1
Yes	1 (0.9)	0	1 (1.2)	
**Metastatic sites,***n* (%)				
Lung	80 (69.0)	20 (64.5)	60 (70.6)	0.6
Liver	18 (15.5)	1 (3.2)	17 (20.0)	0.04
Nodal	32 (27.6)	10 (32.3)	22 (25.9)	0.2
Bone	28 (24.1)	8 (25.8)	20 (23.5)	0.8
Brain	2 (1.7)	1 (3.2)	1 (1.2)	0.5
**First-Line Therapy,***n* (%)				0.3
Sunitinib	66 (56.9)	15(48.4)	51 (60)	
Pazopanib	50 (43.1)	16 (51.6)	34 (40)	
**Line of therapy after TKI,***n* (%)				0.03
>1	33 (28.4)	4 (12.9)	29 (34.1)	

Percentages are expressed on column total. RCC: renal cell carcinoma, ECOG: Eastern Cooperative Oncology Group; IMDC: International Metastatic Renal Cell Carcinoma Database Consortium; TKI: tyrosine kinase inhibitor; *p*: *p*-value.

**Table 3 biomedicines-10-02444-t003:** Patients’ baseline characteristics according to duration of response to TKI in intermediate-poor risk patients.

	All	PFS ≥ 36 Months	PFS < 36 Months	*p*
	*N* = 219	*N* = 35 (16.0%)	*N* = 184 (84.0%)	
**Age**				0.5
Median (range)	65 (37–89)	63 (37–78)	65 (37–89)	
**Gender,***n* (%)				1
Male	157 (71.7)	25 (71.4)	132 (71.7)	
**Histology,***n* (%)				1
Clear-cell RCC	192 (87.7%)	31 (88.6)	161 (87.5)	
**Previous nephrectomy,***n* (%)				0.02
Yes	176 (80.4%)	33 (94.3)	143 (77.7)	
**ECOG,***n* (%)				<0.01
≥1	124 (56.6)	13 (37.1)	11 (60.3)	
**Sarcomatoid feature**				0.2
Yes	13 (5.9)	0	13 (7.1)	
**Metastatic sites,***n* (%)				0.4
Lung	138 (63.0)	20 (57.1)	118 (64.1)	0.3
Liver	33 (15.1)	3 (8.6)	30 (16.3)	0.3
Nodal	85 (38.8)	11 (31.4)	74 (40.2)	0.4
Bone	77 (35.2)	10 (28.6)	67 (36.4)	0.2
Brain	12 (5.5)	0	12 (6.5)	
**First-Line Therapy,***n* (%)				0.6
Sunitinib	134 (61.2)	23 (65.7)	111 (60.3)	
Pazopanib	85 (38.8)	12 (34.3)	73 (39.7)	
**Line of therapy after TKI,***n* (%)				1
>1	69 (31.5)	11 (31.4)	58 (31.5)	

Percentages are expressed on column total. RCC: renal cell carcinoma, ECOG: Eastern Cooperative Oncology Group; IMDC: International Metastatic Renal Cell Carcinoma Database Consortium; TKI: tyrosine kinase inhibitor; *p*: *p*-value.

**Table 4 biomedicines-10-02444-t004:** Univariate and multivariate analysis of the correlation of various clinical-pathological variables with PFS ≥ 36 months in all patients.

Univariate Analysis	Odds Ratio	CI 95%	*p*
**Age**			
>70	0.92	0.53–1.60	0.8
**Gender**			
Male	0.68	0.38–1.21	0.2
**Histology**			
Clear-cell RCC	1.22	0.51–2.89	0.6
**Previous nephrectomy**			
Yes	4.33	1.30–14.39	0.02
**ECOG PS**			
≥1	0.43	0.26–0.72	<0.01
**Sarcomatoid feature**			
Yes	1		
**IMDC score**			
Intermediate-poor	0.52	0.30–0.90	0.02
**Metastatic sites,***n* (%)			
Lung	0.79	0.45–1.37	0.4
Liver	0.3	0.10–0.88	0.02
Nodal	0.84	0.47–1.49	0.5
Bone	0.78	0.43–1.43	0.4
Others	1.46	0.82–1.59	0.2
**Multivariate analysis**	**Odds Ratio**	**CI 95%**	** *p* **
**Previous nephrectomy**			
Yes	3.42	1.00–11.63	0.05
**ECOG**			
≥1	0.5	0.30–0.85	0.01
**IMDC score**			
Intermediate-poor	0.64	0.36–1.14	0.1
**Metastatic sites,***n* (%)			
Liver	0.32	0.11–0.93	0.04

RCC: renal cell carcinoma, ECOG PS: Eastern Cooperative Oncology Group Performance Status; IMDC: International Metastatic Renal Cell Carcinoma Database Consortium; TKI: tyrosine kinase inhibitor; CI: confidence interval; *p*: *p*-value.

**Table 5 biomedicines-10-02444-t005:** Univariate and multivariate analysis of the relationships between various clinical-pathological variables with PFS ≥ 36 months in good-risk patients.

Univariate Analysis	Odds Ratio	CI 95%	*p*
**Age**			
>70	1.5	0.66–3.43	0.3
**Gender**			
Male	0.43	0.18–1.02	0.05
**Histology**			
Clear-cell RCC	1.39	0.36–5.34	0.6
**Previous nephrectomy**			
Yes	1.87	0.21–16.71	0.6
**ECOG**			
≥1	0.54	0.22–1.29	0.2
**Sarcomatoid feature**			
Yes	1		
**Metastatic sites,***n* (%)			
Lung	0.75	0.32–1.81	0.5
Liver	0.13	0.02–1.05	0.05
Nodal	1.37	0.56–3.34	0.5
Bone	1.13	0.44–2.92	0.8
Others	1.19	0.50–2.85	0.7
**Multivariate analysis**	**Odds Ratio**	**CI 95%**	** *p* **
**Gender**			
Male	0.38	0.15–0.95	0.04
**Metastatic sites,***n* (%)			
Liver	0.32	0.11–0.93	0.04

RCC: renal cell carcinoma, ECOG PS: Eastern Cooperative Oncology Group Performance Status; IMDC: International Metastatic Renal Cell Carcinoma Database Consortium; TKI: tyrosine kinase inhibitor; CI: confidence interval; *p*: *p*-value.

**Table 6 biomedicines-10-02444-t006:** Univariate and multivariate analysis of the relationships between various clinical-pathological variables with PFS ≥ 36 months in intermediate-poor risk patients.

Univariate Analysis	Odds Ratio	CI 95%	*p*
**Age**			
>70	0.53	0.23–1.19	0.1
**Gender**			
Male	0.98	0.44–2.19	0.9
**Histology**			
Clear-cell RCC	1.11	0.36–3.42	0.9
**Previous nephrectomy**			
Yes	4.73	1.01–20.55	0.04
**ECOG**			
≥1	0.39	0.18–0.82	0.01
**Sarcomatoid feature**			
Yes	1		
**Metastatic sites,** *n* (%)			
Lung	0.74	0.36–1.55	0.4
Liver	0.48	0.14–1.67	0.2
Nodal	0.68	0.31–1.47	0.3
Bone	0.7	0.32–1.54	0.4
Others	1.81	0.82–3.95	0.1
**Multivariate analysis**	**Odds Ratio**	**CI 95%**	** *p* **
**Previous nephrectomy**			
Yes	3.98	0.90–17.56	0.1
**ECOG**			
≥1	0.44	0.21–0.94	0.03

RCC: renal cell carcinoma, ECOG PS: Eastern Cooperative Oncology Group Performance Status; IMDC: International Metastatic Renal Cell Carcinoma Database Consortium; TKI: tyrosine kinase inhibitor; CI: confidence interval; *p*: *p*-value.

## Data Availability

The data used to support the findings of this study are available from the corresponding author upon request.

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
