# Peer review of "Long-Term Response to Tyrosine Kinase Inhibitors for Metastatic Renal Cell Carcinoma"

_biomedicines, 2022, doi:10.3390/biomedicines10102444_

Round 1
Reviewer 1 Report
The paper is about a retrospective analysis of long-term response to tyrosine kinase Inhibitors for metastatic renal cell carcinoma. Of 335 patients treated with sunitinib or pazopanib 66 patients showed long-term (≥ 36 months) progression free survival. Previous nephrectomy, absence absence of liver metastasis, intermediate-poor IMDC score (not in multivariate analysis) and ECOG≥1 are significantly more often seen in long-term survivors.
The following comments should be answered:
It is of interest that for the subgroups of favourable and intermediate-poor IDMC score patients different characteristics are significant and that the IDMC score is not significant in multivariate analysis. What is the explanation for this? Has this any impact for a classification of a risk score or the treatment in the future even with other drugs or drug combinations?
It seems obvious that in favorable risk patients in the short responder group more patients receive more than one TKI. If there is a better PFS, why should treatment be changed as often in the long responder group. The same question can be asked for RR and DCR in the two different risk groups. If you go for 5-year PFS as a landmark type then PFS will be better, but number of patients will be smaller. What was the reaon to use 36 months as a cut-off?
What is an explanation why gender is significant in low score patients with PFS ≥36 months and ECOG≥1 only in patienst with intermediate-poor risk but previous nephrectomy not?
Even if in the favorable risk population, combination therapies did not show a 262 significant advantage in terms of OS over TKI alone the question is if in a group of patients no liver metastasis in this group there might be an advantage of a combination therapy for this risk group?
Minor issue:
Methods:
line 105 and 106 reads that only patients with pzopanib were under the 66 long-term survivors. In table 1, this is not the case. Should be updated.
Author Response
The paper is about a retrospective analysis of long-term response to tyrosine kinase Inhibitors for metastatic renal cell carcinoma. Of 335 patients treated with sunitinib or pazopanib 66 patients showed long-term (≥ 36 months) progressionfree survival. Previous nephrectomy, absence of liver metastasis, intermediate-poor IMDC score (not inmultivariate analysis) and ECOG≥1 are significantly more often seen in long-term survivors. The following comments should be answered:
Point 1. It is of interest that forthe subgroups of favourable and intermediate-poor IDMC score patients different characteristics are significant and that the IDMC score is not significant in multivariate analysis. What is the explanation for this? Has this any impact for a classification of a risk score or the treatment in the future even with other drugs or drug combinations?
Response 1. Thanks for your comment. Although risk classification is currently the cornerstone in the choice of treatment in patients with mRCC, some parameters, such as burden disease, or metastaticsite are not considered among the different risk actors.We think that these elements could be significant in the therapeutic choice especially considering the different possibilities currently available (TKI, IO-TKI, IO-IO).
Point 2. It seems obvious that in favorable risk patients in the short responder group more patients receive more than one TKI. If there is a better PFS, why should treatment be changed as often in the long responder group. The same question can beasked for RR and DCR in the two different risk groups. If you go for 5-year PFS as a landmark type then PFS will bebetter, but number of patients will be smaller. What was the reaon to use 36 months as a cut-off?
Response 2. Thank you, this comment is very relevant. Although the choice of the cut off is arbitrary, the purpose of considering such a long PFS was to select, at most, theTKI responders. However, recognizing it as a limit, we have added it to thelimitations of the study.
Point 3. What is an explanation why gender is significant in low score patients with PFS ≥36 months and ECOG≥1 only in patienst with intermediate-poor risk but previous nephrectomy not?
Response 3. As regards gender, the data is consistent with the results of the literature where the female gender seems to be correlated with better overall survival, recurrence rates and cancer specific survival than men with RCC (Zeynep et al., Urology,2021). This information is now reported in the discussion section. Regarding performance status, we think that performance status affects much more than other variables on the prognosis of intermediate/poor riskpopulation. Instead, contrary to what emerged in the univariate analysis, the lack of significance regarding nephrectomy in the intermediate-poor risk group is consistent with data literature. Data regarding cytoreductive nephrectomy did not show benefits in patients with poor risk but only in favorable risk and selected intermediate risk patients.
Point 4. Even if in the favorable risk population, combination therapies did not show a 262 significant advantage in terms of OSover TKI alone the question is if in a group of patients no liver metastasis in this group there might be an advantage of acombination therapy for this risk group?
Response 4. Currently, the combination of IO and TKI is the first-line recommended treatment option for patients with mRCC,although the OS benefit appears negligible in favorable risk patients. In these patients, carefully selected,TKI alone may still be an option; however, in our opinion, combination treatment should be recommended in patients with a high burden of disease even in the absence of liver metastases, in young patients with rapidly evolving disease and in those with brain metastasis. We modified the discussion accordingly.
Minor issue: Methods: line 105 and 106 reads that only patients with pazopanib were under the 66 long-term survivors. In table 1, this is notthe case. Should be updated.
Thank you for pointing out the errorwhich has been corrected.

Reviewer 2 Report
The authors present to us a good paper reflecting treatment of metastatic renal cell carcinoma with the use of TKI.
The paper reflects most of the actual practice for the last years and can be very useful to support some clinic decisions.
There are some minor changes in order to accept for publishing.
I want to stress out some positive points and some changes to be made:
Introduction
- there is a good summary and justification to the objective of the paper.
- it would be nice, if possible, to have a real world characterisation of therapeutics in this setting, because we know that TKI are still commonly used despite guidelines
Material and Methods
- well constructed
- recommend some changes in english forms line 72 - where it is written "... malignancy or previously treatment" should be "malignancy or previously treated".
- "All patients received target therapy with sunitinib 50 mg for four consecutively weeks and two weeks stop or pazopanib 800 mg one a day continuatively up to intolerance or progression disease" line 74-75 --> none of the patients did sunitinib in 2 to1 scheme?
Results
this section is logically sequenced as the division in short term and long term responders is the main line in this work
the graphic named figure 3 has a minor typo in \landmark
Discussion
Despite corroborating data from your study with some other papers, I thin that some consideration about the role of previous nefphrectomy for the response time.
Despite being a good paper, reflecting real world data and most common practice, there are some concerns about this paper:
- retrospective design
- some excluded patients by missing data can change results
- lack of information about duration of treatment
Author Response
Review Report 2
The authors present to us a good paper reflecting treatment of metastatic renal cell carcinoma with the use of TKI. The paper reflects most of the actual practice for the last years and can be very useful to support some clinic decisions.
There are some minor changes in order to accept for publishing. I want to stress out some positive points and some changes to be made:
Point 1. Introduction-there is a good summary and justification to the objective of the paper.-it would be nice, if possible, to have a real-world characterization of therapeutics in this setting, because we know thatTKI are still commonly used despite guidelines
Response 1. Thank you, this is an optimal observation and an interesting input for future research. Indeed, now we don’t able to exactly evaluate the percentage of patients who are receiving TKI as first line.
Point 2. Material and Methods-well constructed-recommend some changes in english forms line 72-where it is written "...malignancy or previously treatment"should be "malignancy or previously treated".
Response 2. Thanks, the error has been corrected.
Point 3. -"All patients received target therapy with sunitinib 50 mg for four consecutively weeks and two weeks stop or pazopanib 800 mg one a day continuatively up to intolerance or progression disease" line 74-75--> none of the patients did sunitinib in 2 to1 scheme?
Response 3. This is an interesting observation. In our study we reported only the initial treatment schedules, whereas data regarding the adjustment of doses during treatment are missing. This limitation has been added in the discussion.
Point 4. Results this section is logically sequenced as the division in short term and long-term responders is the main line in this work the graphic named figure 3 has aminor typo in\landmark
Response 4. Thank you, typo has been corrected.
Point 5. Discussion
Despite corroborating data from your study with some other papers, I think that some consideration about the role of previous nefphrectomy for the response time.
Response 5. This comment is very interesting as the role of surgery in metastatic RCCis still highly debated. However, in our study we refer to surgery in a non-metastatic setting, where the time to disease recurrence is a known prognostic factor, for which, we did not prefer to argue further.
Point 6. Despite being a good paper, reflecting real world data and most common practice, there are some concerns about thispaper:-retrospective design-some excluded patients by missing data can change results-lack o of information about duration of treatment.
Response 6. Thank you. In agreement with your comment, we added some limitation not previous reported.
